# Comparison of Disinfection By-Product Formation and Distribution during Breakpoint Chlorination and Chlorine-Based Disinfection in Drinking Water

**Dávid Stefán** [1,2]**, Judit Balogh** [3]**, Gyula Záray** [1,4] **and Márta Vargha** [2,*]

1 Hevesy György PhD School of Chemistry, Institute of Chemistry, Eötvös Loránd University, Pázmány Péter Street 2, 1117 Budapest, Hungary; stefan.david@nnk.gov.hu (D.S.); zaray@ludens.elte.hu (G.Z.)
2 Department of Public Health Laboratories, National Public Health Center, Albert Flórián Street 2-6, 1097 Budapest, Hungary
3 Department of Inorganic and Analytical Chemistry, Budapest University of Technology and Economics, Műegyetem Rakpart 3, 1111 Budapest, Hungary; juditbalogh21@gmail.com
4 Danube Research Institute, Centre for Ecological Research, Hungarian Academy of Sciences, Karolina Street 29-31, 1113 Budapest, Hungary
* Correspondence: vargha.marta@nnk.gov.hu

**Abstract:** Breakpoint chlorination (BC) and disinfection with chlorine-based disinfectant are widely used procedures in drinking water production. Both involve dosing chlorine into the raw water, where it can react with organic compounds, forming disinfection by-products (DBPs) of health concern. However, technological parameters (e.g., contact time, chlorine dosage, and bromide to residual free chlorine ratio) of the two chlorination procedures are different, which can lead to differences in DBP formation. To better understand this, a year-long sampling campaign was carried out at three waterworks in Hungary, where both BC and chlorine disinfection are used. To confirm the results of the field sampling, bench-scale experiments were carried out, investigating the impact of (a) bromide concentration in raw water, (b) residual free chlorine (bromide to residual chlorine ratio), and (c) contact time on DBP formation. The measured DBPs were trihalomethanes (THMs), haloacetic acids (HAAs), haloacetonitriles (HANs), and chlorate. During BC, the DBPs were formed in higher concentration, with the exception of one waterwork having elevated bromide content in the raw water. Bromine substitution factors (BSFs) were significantly higher during disinfection than BC in both field and laboratory experiments. After BC, the chlorate concentration range was 0.15–1.1 mg/L, and 96% of the samples exceeded the European Union (EU) parametric value (0.25 mg/L), whereas disinfection contributed only slightly. Granular activated carbon (GAC) filters used to remove DBPs in waterworks were exhausted after 6–8 months of use, first for those chlorinated THMs, which are generated predominantly during BC. The biological activity of the filters started to increase after 3–6 months of operation. This activity helps to remove the biodegradable compounds, such as disubstituted haloacetic acid (DHAAs) and HANs, even if the adsorption capacity of the GAC filters are low.

**Keywords:** breakpoint chlorination; bromine substitution factors; chlorate; disinfection; disinfection by-products; aging of GAC adsorbents





## 1. Introduction

Water chlorination is an effective and cost-efficient technology for the disinfection of drinking water. When using chlorine or hypochlorite, the hypochlorite ion and hypochlorous acid are the species responsible for the disinfectant effect. However, the reaction of natural organic matter and the added chlorine-based disinfectant leads to the formation of various halogen-containing disinfection by-products (DBPs) detrimental to health [1,2]. Of the over 1000 known DBPs [3], the most abundant are trihalomethanes (THMs) and

haloacetic acids (HAAs), but nitrogen-containing DBPs such as haloacetonitriles (HANs), halonitromethanes (HNMs), or N-nitrozodimethylamine (NDMA) are also produced in lower concentrations [4–6].

The potential health impact of these components is well known. Long-term exposure to THMs can be harmful to the liver, kidney, and central nervous system [7]. Chloroform and bromo-dichloromethane are listed as Group 2B carcinogen (possibly carcinogenic to humans) compounds by the International Agency for Research and Cancer (IARC) [8]. The carcinogenicity of HAAs is also under investigation: dichloroacetic acid, trichloroacetic acid, bromo-chloroacetic acid, and dibromoacetic acid are also classified in Group 2B [8]. Toxicological reviews by the United States Environmental Protection Agency (US EPA) indicate that both dichloroacetic acid and trichloroacetic acid primarily damage the liver [9,10]. Dichloroacetic acid also has neurotoxic effects. Chronic exposure to chlorate and chlorite can cause anemia and damage the nervous system in young children [11], but they are not classified by IARC because of insufficient toxicological data.

The recently adopted recast European Union drinking water directive (DWD) places a stronger focus on DBPs than previously [12]. In addition to THMs, HAAs and inorganic DBPs chlorite and chlorate were introduced into the list of parameters to be monitored and controlled. Parametric values are 100 and 60 μg/L for THMs and HAAs, respectively, and 0.25 mg/L for chlorite and chlorate [12]. DWD does not address nitrogen-containing DBPs. However, several studies have shown that many of the unregulated DBPs can be more genotoxic or cytotoxic then those currently regulated [13,14]. Although nitrogen-containing DBPs typically occur at lower concentrations than THMs and HAAs, their significance may be offset by their greater toxicity [15,16].

Chlorine dosing has another, less common role in drinking water treatment beside disinfection: ammonium removal by breakpoint chlorination (BC). Some water sources contain elevated concentrations of ammoniums ion either from anthropogenic or—as is the case with groundwater from deep aquifers in Hungary—natural sources [17]. Although ammonium does not have an adverse health effect, it can be oxidized to nitrite in the water supply or distribution system under certain conditions, which can cause methemoglobinemia in infants [18]. Chlorine-based reagents oxidize ammonium to nitrogen through chloramine intermediates. The excess reagent is removed by filtration on granular activated carbon (GAC) or biological activated carbon (BAC) adsorbents. A second dose of chlorine is usually applied after ammonium removal for disinfection, because the disinfection efficiency of residual combined chlorine produced during BC is lower than that of active chlorine.

GAC or BAC filtration after BC eliminates residual free chlorine and DBPs to varying extents. In GAC, the dominant removal process is adsorption, whereas in BAC it is biodegradation. Both techniques can lower the cytotoxicity and genotoxicity of the treated water [19,20]. However, during the aging of GAC sorbents, the adsorption efficiency of certain DBPs, such as THMs, can decrease rapidly [21]. GAC/BAC filtration after disinfection is rare, as it removes residual chlorine and reduces disinfection efficiency during distribution. For the removal of water contaminants, other alternative adsorbents, e.g., carbon nanotubes [22], carbon quantum dots [23], or addition of melamine [24], can be used.

The factors influencing organic DBP formation during disinfection are well characterized. The most crucial are: (a) the amount and the quality of the organic precursors [25]; (b) the type and the dosage of the disinfectant used [26,27]; (c) bromide concentration of raw water [25,27]; (d) the ratio of residual free chlorine to bromide ion ($Cl_2/Br^-$) [25,27]; and (e) physical–chemical parameters (temperature, pH etc.) [25–27].

However, the two chlorination technologies differ in several parameters, many of which have an impact on DBP formation, such as contact time, concentration of residual free chlorine, and $Cl_2/Br^-$ ratio or water temperature (Table S1). BC is characterized by short and controlled contact time (minutes) with a large dose of chlorine (1.0–10.0 mg $Cl_2$/L) and high concentration of residual free chlorine. During disinfection, the contact time can

vary from hours to days depending on the residence time in the distribution system. The chlorine dosage is much lower (0.3–1.0 mg $Cl_2$/L), and the concentration of residual free chlorine can be adjusted more precisely. Due to the differences, the relative importance of these parameters and their effects on the DBP formation are not necessarily the same. The two chlorination processes applied consecutively are expected to lead to enhanced DBP formation.

Of the inorganic DBPs addressed by the DWD, chlorite is mainly generated from chlorine dioxide, whereas chlorate is mainly produced during the decomposition of hypochlorite ions during storage [28]. The higher dose of hypochlorite used for BC is expected to lead to increased concentration of chlorate, but limited information is available due to the lack of prior regulation or monitoring requirement.

The primary objective of this study was to assess whether DBPs from BC pose increased risk to human health in drinking water production compared to chlorine-based disinfection. Although the process of BC in wastewater is well studied, limited information is available about the process in drinking water. To the best of our knowledge, this is the first study investigating the formation and distribution of organic and inorganic DBPs (THMs, HAAs, HANs, and chlorate) during BC and chlorine-based disinfection simultaneously. For this purpose, two sets of experiments were performed: (1) a field investigation at drinking water treatment plants using both procedures; and (2) bench-scale experiments with artificial raw water. Since the primary tool for DBP removal after BC is GAC filtration, the impact of the aging of GAC sorbents on the quantity and distribution of DBPs formed during BC was investigated.

## 2. Materials and Methods

### 2.1. Field Sampling

A one-year sampling campaign was carried out at 3 drinking water treatment plants (DWTPs) in different parts of Hungary. The DWTPs are indicated below by Roman numerals (I–III). Source waters were abstracted from protected deep aquifers in each DWTP. Sodium hypochlorite was applied to remove the relatively high ammonium concentration of the raw waters (0.80–1.2 mg/L). BC was followed by GAC filtration and disinfection using also sodium hypochlorite. The schematic flow chart of the treatment technologies is shown in Figure 1. Fresh GAC adsorbent was installed prior to the sampling campaign at every DWTP. Samplings were carried out 1, 2, 3, 4, 6, 8, 10, and 12 months after GAC replacement between October 2019 and October 2020. Samples were collected from (a) raw water, (b) treated water after breakpoint reagent dosing, (c) after GAC filters, (d) finished water (ex-waterworks), and (e) tap water at the point of consumption (Figure 1).

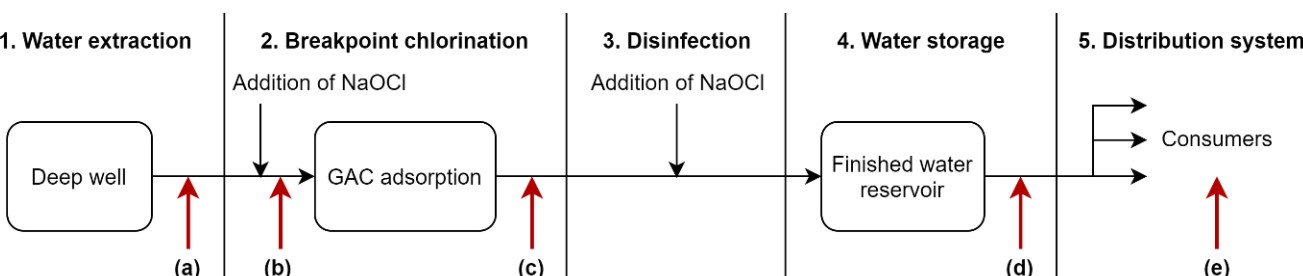

**Figure 1.** Schematic flow chart of the investigated DWTPs with the sampling points (**a**–**e**) indicated.

### 2.2. Bench-Scale Experiments

2.2.1. Raw Water Composition

Synthetic raw water was prepared from deionized water containing the following components: sodium ions (123 mg/L), magnesium ions (19 mg/L), calcium ions (80 mg/L), chloride ions (197 mg/L), DOC (humic acid, 0.60 mg/L). Ammonium ions (3.0 mg/L) were added for BC experiments. The pH was adjusted to 7.5 ± 0.1 by hydrogen carbonate buffer. The synthetic raw water simulates the usual composition of groundwaters used as a

drinking water source in Hungary. Bromide ions were added in varying concentration (see below). Sodium hypochlorite solution (15 g $Cl_2$/L) was used for both breakpoint oxidation and disinfection. The investigated variables were (1) residual free chlorine, (2) bromide concentration of the raw water, and (3) contact time. Experiments were performed at room temperature (20–25 °C)

### 2.2.2. Breakpoint Chlorination

Three different $Cl_2$ dosages were applied to a residual free chlorine concentration of 1.5, 3.0, and 6.0 mg $Cl_2$/L after breakpoint reaction. The impact of each $Cl_2$ dosage was assessed at three bromide levels (0.1, 0.2, and 0.4 mg/L); thus, a total of 9 different treatments were tested. The applied bromide concentration range represents the composition of Hungarian groundwaters used as a drinking water source. The synthetic raw water and the breakpoint reagent were mixed and sampled after 15 min reaction time.

To assess the impact of contact time, bromide concentration was set to 0.17 mg/L in the synthetic water and final residual free chlorine concentration was adjusted to 2.0 mg $Cl_2$/L. Samples were collected 0, 3, 6, 9, 12, 16, 20, 25, and 30 min after addition of the reagent.

### 2.2.3. Disinfection

Disinfection experiments were carried out using the same synthetic raw water without ammonium. Bromide was added in three different concentrations as described above. Residual free chlorine concentration was set to 0.3, 0.6, and 1.2 mg $Cl_2$/L. All nine combinations of chlorine dosage levels and bromide concentrations were tested. Solutions were mixed and kept in the dark to minimize the photodegradation of hypochlorite. Sampling was carried out after 24 h, which represents an average residence time in the water distribution systems.

### 2.3. Analytical Methods

THMs, HAAs, and HANs were measured as organic DBPs and chlorate as inorganic DBP. Other relevant chemical parameters of the water samples were also analyzed. All the investigated parameters with the applied analytical methods and the corresponding detection limits are listed in Table 1.

Free and combined chlorine levels were determined at the sampling sites using N,N-diethyl-1,4-phenylenediamine (DPD) colorimetric titration according to the ISO 7393-1:1985 method [29]. The temperature, pH, and conductivity of raw water were also measured on site.

**Table 1.** Analyzed parameters during field sampling and bench-scale experiments.

|  | Parameters | Measurement Principle | Measurement Methods | Limit of Quantitation (LOQ) |
|---|---|---|---|---|
| On site | temperature, pH, conductivity | - | - | - |
|  | free and combined chlorine | DPD colorimetric titration | ISO 7393-1:1985 [29] | 0.030 mg $Cl_2$/L |
| Basic parameters | DOC | combustion + IR detection | UNE EN 1484:1998 [30] | 0.50 mg/L |
|  | ammonium | photometric | ISO 7150-1:1992 [31] | 0.020 mg/L |
|  | chloride, bromide, nitrite, nitrate | IC + conductivity detection | ISO 10304-1:2007 [32] | 2.0, 0.050, 0.030, and 0.50 mg/L, respectively |
| Organic DBPs | 4 THMs: chloroform, bromo-dichloro-methane (BDCM), dibromo-chloromethane (DBCM), bromoform | Purge & Trap-GC-MS | - | 0.10 μg/L |

| | Parameters | Measurement Principle | Measurement Methods | Limit of Quantitation (LOQ) |
|---|---|---|---|---|
| | 9 HAAs: monochloroacetic acid (MCAA), monobromoacetic acid (MBAA), dichloroacetic acid (DCAA), trichloroacetic acid (TCAA), bromo-chloroacetic acid (BCAA), dibromoacetic acid (DBAA), bromo-dichloroacetic acid (BDCAA), dibromo-chloroacetic acid (DBCAA), tribromoacetic acid (TBAA) | Liquid-liquid extraction + derivatization + GC-MS | EPA 552.3 [33] with slight changes | 0.50 µg/L (except MCAA: 1.0 µg/L) |
| | 3 HANs: dichloroacetonitrile (DCAN), bromo-chloroacetonitrile (BCAN), dibromoacetonitrile (DBAN) | Liquid-liquid extraction + GC-ECD | EPA 551.1 [34] with minor modification | 0.30 µg/L |
| Inorganic DBPs | Chlorate | IC + cond. detection | ISO 10304-4:2000 [35] | 0.050 mg/L |

Ammonium was determined according to ISO 7150-1:1992 [31]. Briefly, the ammonium was converted to monochloramine with sodium-dichloroisocyanurate. Then the monochloramine reacted with sodium-salicylate in the presence of nitroprusside ions forming a blue product. The absorbance of the solution was measured at 655 nm. Dissolved organic carbon (DOC) was measured after filtration and acidification of the samples using a vario TOC cube (Elementar, Langenselbold, Germany). The anions (chloride, bromide, nitrite, nitrate, and chlorate) were measured with an ICS 5000+ ion chromatograph (Dionex, Sunnyvale, California, United States). After direct injection on a Dionex IonPacTM AS11-HC (Thermo Scientific, Waltham, Massachusetts, United States) anion exchange column, the detection was carried out with a conductivity detector. Analysis was started within 24 h of sample collection to avoid the transformation of the analytes.

THMs were measured with gas chromatograph-mass spectrometer coupled with an online Purge & Trap sample preparation system (P & T-GC-MS). Sample volumes of 40 mL were added to 40 mL EPA glass vials and spiked with internal standard solution (1,2-dichlorobenzene-d4). The samples were placed into the autosampler of the P&T instrument (AquaTek 100, Teledyne Tekmar, Mason, Ohio, United States). The thermostat was adjusted to 50 °C and the samples were purged for 8 min with $N_2$ gas. After trapping at room temperature, the activated carbon trap was heated immediately to 250 °C for 2 min and the samples were transferred with helium carrier gas to the gas chromatograph (7890B, Agilent Technologies, Santa Clara, California, United States). The compounds were separated on a 20 m × 0.18 mm × 1.0 µm RTX-VMS column (Restek Co., Bellefonte, Pennsylvania, United States). The temperature program was the following: initial 35 °C for 5 min, 25 °C/min to 110 °C, 40 °C/min to 240 °C held for 2 min. THMs were detected with a mass spectrometer (7000C, Agilent Technologies, Santa Clara, California, United States) in SIM mode and quantified by internal standard calibration.

The determination of 9 HAAs was carried out according to the USEPA 552.3 (2003) standard [33] with a GC-MS system (HP 6890 and HP5973A, Hewlett Packard, Santa Clara, California, United States). Briefly, internal standard (1,2,3-trichloropropane), sodium sulfate and concentrated sulfuric acid were added to water samples and extracted with methyl tert-butyl ether (MTBE) in acidic media (pH < 2). This was followed by derivatization with acidic methanol solution. The solution was heated to 50 °C for 2 h. After cooling, samples were extracted by sodium sulfate and sodium bicarbonate solutions. A 1 mL aliquot of the organic phase was sealed in a 2 mL amber glass vial and stored at −20 °C until analysis. The standard method was slightly modified: 50 mL water sample was

extracted and 30 m × 0.25 mm × 0.25 μm HP-5ms UI column (Agilent Technologies, Santa Clara, California, United States) was used for GC separation. The temperature program was also modified: 35 °C for 15 min, heating at 2.5 °C/min to 60 °C, 10 °C/min to 80 °C, and finally 20 °C/min to 210 °C held for 3 min. A mass spectrometer was used for detection in SIM mode.

HANs were analyzed with a GC-ECD system (HP 5890, Hewlett Packard, Santa Clara, California, United States) according to the US EPA 551.1 (1995) standard [34] with slight changes. Briefly, internal standard (1,2,3-trichloropropane), sodium sulphate, and $HPO_4^{2-}/H_2PO_4^-$ buffer were added to 50 mL of the sample and extracted by 3 mL of MTBE. A 1 mL aliquot of the organic phase was sealed in a 2 mL amber glass vial and stored at −20 °C until analysis. The separation was carried out on a 30 m × 0.25 mm × 0.25 μm RH-5ms+ column. Nitrogen was used as carrier gas and the head pressure was set to 15 psi. The temperature was the following: initial 35 °C held for 6 min, 10 °C/min to 100 °C, and 40 °C/min to 220 °C held for 3 min.

### 2.4. Data Analysis

The amount of DBPs originating from the chlorination processes in field sampling was calculated according to Equations (1) and (2).

$$\text{DBPs produced during BC } (\mu g/L) = \text{conc. at (b)} - \text{conc. at (a)} \tag{1}$$

$$\text{DBPs produced during disinf.} (\mu g/L) = \text{conc. at (e)} - \text{conc. at (c)} \tag{2}$$

Equations (3)–(5) were used for the calculation of the mass concentration (μg/L) of disubstituted HAAs (DHAAs), trisubstituted HAAs (THAAs), and disubstituted HANs (DHANs), respectively.

$$c_{DHAAs} = c_{DCAA} + c_{BCAA} + c_{DBAA} \tag{3}$$

$$c_{THAAs} = c_{TCAA} + c_{BDCAA} + c_{DBCAA} + c_{TBAA} \tag{4}$$

$$c_{DHANs} = c_{DCAN} + c_{BCAN} + c_{DBAN} \tag{5}$$

The degree of bromination during DBP formation was calculated using bromine substitution factor (BSF) [36]. BSF is defined as a ratio of molar concentration of bromine incorporated into a given class of DBP to the total molar concentration of chlorine and bromine in the given class. As an example, BSF for THMs can be calculated according to Equation (6). BSFs are unitless values.

$$\text{BSF (THMs)} = \frac{[CHCl_2Br] + 2[CHClBr_2] + 3[CHBr_3]}{3[[CHCl_3] + [CHCl_2Br] + [CHClBr_2] + [CHBr_3]]} \tag{6}$$

Similar equations were used for DHAAs, THAAs, and DHANs. BSF always ranges from 0 to 1, thus allowing the comparison of BSF over different class of DBPs.

The DBP removal efficiency of the GAC adsorbents were investigated using concentrations measured at sampling points (b) and (c). The efficiency is given as percent (%) according to Equation (7).

$$\text{Efficiency } (\%) = \frac{\text{conc. at (b)} - \text{conc. at (c)}}{\text{conc. at (b)}} \times 100 \tag{7}$$

If the concentration of a DBP was under the LOQ at sampling point (c), the LOQ value was applied for the calculation, thus obtaining the minimum efficiency of the sorbents.

The descriptive statistics and the visualization of the results were carried out with Microsoft Excel™ software, version 2109.

## 3. Results and Discussion

### 3.1. Field Sampling

3.1.1. Raw Water and Technology Parameters

Characteristics of the raw water at each sampling site are summarized in Table 2. Only slight differences were observed between the sites in raw water temperature, pH, and conductivity. The ammonium concentration varied between 0.84 and 1.2 mg/L. $Br^-$ and TOC concentrations were more diverse: the average $Br^-$ concentration at waterwork I was 0.17 mg/L, whereas it was under the detection limit (0.05 mg/L) at the other two locations. DOC concentration was higher at waterworks II and III (2.2 and 2.3 mg/L, respectively) than at waterwork I (1.0 mg/L). Both parameters are important as precursors of DBPs. As expected, the composition of the groundwaters originating from deep aquifers was constant over time during the year-long sampling campaign. The chemical composition of raw water at site II and III was very similar. Residual free chlorine values vary uniformly around 3 mg/L after BC and 0.6 mg/L after disinfection at every sampling site. Free and combined chlorine concentrations measured after BC indicate that the breakpoint reaction was complete at every DWTP.

**Table 2.** The means and standard deviations (SDs) of raw water and technology parameters in the investigated waterworks I–III.

| Parameters | | Waterwork I | | Waterwork II | | Waterwork III | |
|---|---|---|---|---|---|---|---|
| | | Mean | SD | Mean | SD | Mean | SD |
| Raw water parameters ($n = 8$) | Temperature (°C) | 17.0 | 0.1 | 18.8 | 0.7 | 15.2 | 0.5 |
| | pH | 7.97 | 0.08 | 7.67 | 0.10 | 7.75 | 0.08 |
| | Cond (μS/cm) | 812 | 7 | 711 | 10 | 643 | 40 |
| | $NH_3$-N (mg/L) | 1.2 | 0.04 | 1.0 | 0.03 | 0.84 | 0.11 |
| | $Br^-$ (mg/L) | 0.17 | 0.02 | <0.05 | - | <0.05 | - |
| | DOC (mg/L) | 1.0 | 0.2 | 2.3 | 0.2 | 2.2 | 0.2 |
| Technology parameters ($n = 8$) | Res. free chlorine at breakpoint (mg $Cl_2$/L) | 3.0 | 0.91 | 2.9 | 0.95 | 2.9 | 0.88 |
| | Res. combined chlorine at disinfection (mg $Cl_2$/L) | 0.60 | 0.20 | 0.58 | 0.11 | 0.70 | 0.24 |

3.1.2. DBP Formation

Organic DBPs

Generally, THMs were generated in the highest concentrations, followed by HAAs and HANs. At the consumers' tap (sampling point (e)), measured concentration ranges were 10.3–34.0 μg/L, 5.5–20.6 μg/L, and 1.5–4.4 μg/L in DWTPs I–III, respectively. None of the measured organic DBP concentrations exceeded the EU parametric values at any point of the water treatment [12].

The observed concentrations varied between sites and between BC and disinfection. At site I, every DBP class was generated in higher concentration during disinfection, than during BC (Figure 2). The difference was within the margin of error in the case of HAAs and HANs (4.8 and 4.6% respectively), but much higher (48.5%) and significant ($\alpha = 0.05$) for THMs. At sites II and III, the formation of THMs and HAAs was at least 100%, and of HANs was at least 50% higher during BC, than after disinfection. As the water treatment technologies at the sites are almost identical, the observed differences are probably associated with the raw water composition.

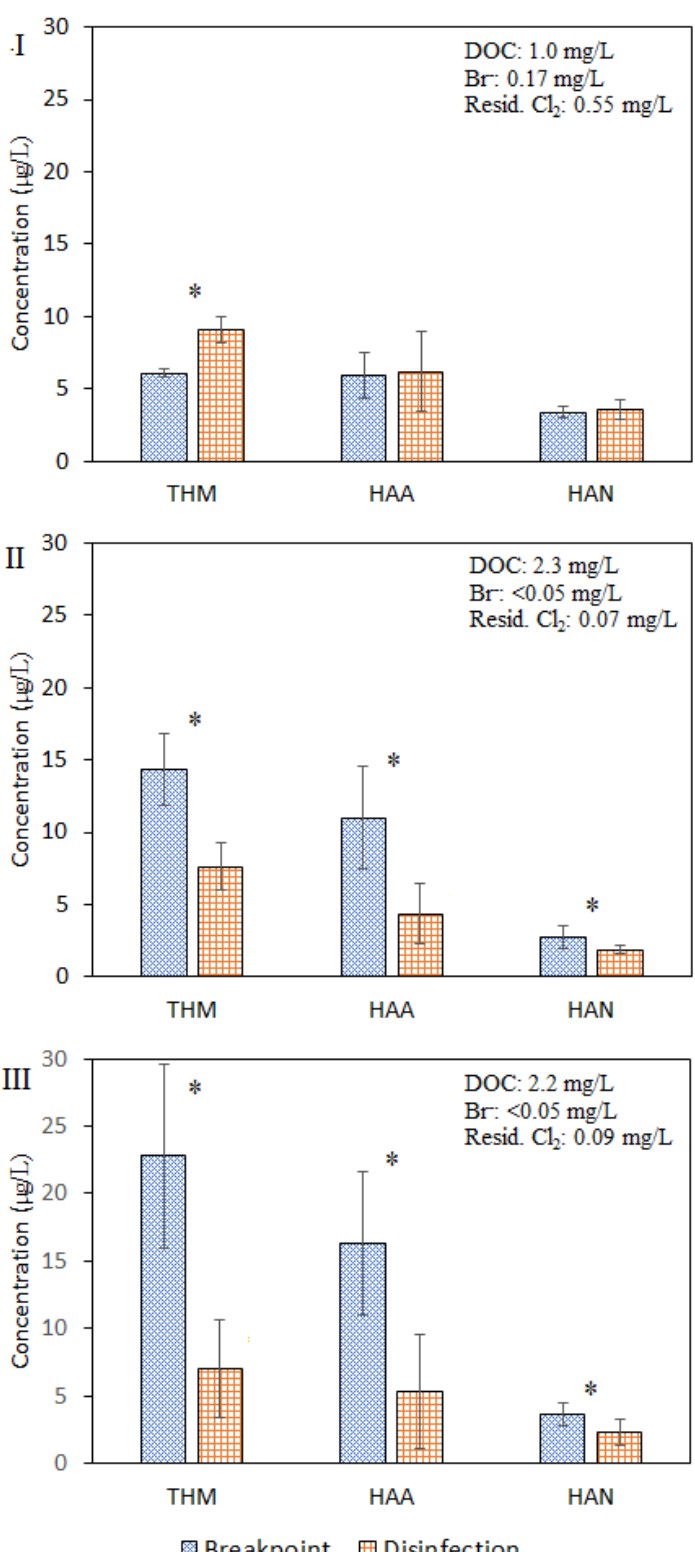

**Figure 2.** Concentration of DBPs generated during BC and disinfection at waterworks **I–III** (* significant difference between BC and disinfection at significance level 0.05).

Comparing the sites, during BC, higher DBP formation was observed at II and III where DOC concentrations in the raw water are higher. During disinfection, the effect was reversed: at site I, where DOC is much lower, concentrations of the formed DBPs exceeded the values measured at the other two sites (Figure 2). The estimated residence time in the

distribution system, represented by sampling point (e) of site I, is lower (24–36 h) than at the sampling point (e) of sites II and III (36–72 h). Accordingly, the average residual free chlorine concentrations at sites II and III were considerably lower than at site I (0.07, 0.09 and 0.55 mg $Cl_2$/L, respectively), implying further DBP formation potential at the latter in the case of longer residence time in the distribution system. A possible explanation for this phenomenon is the different $Br^-$ concentration of raw waters. $Br^-$ can be readily oxidized to HOBr, $Br_2$, or BrCl in the presence of free chlorine [37,38]. An elevated $Br^-$ concentration with the lower chlorine dosage applied during disinfection results in a high HOBr/HOCl ratio. Oxidized bromine compounds are strong halogenating agents, attacking more sites in DBP precursors and reacting with them faster than HOCl [39]. This phenomenon was most likely responsible for the relatively greater DBP formation at disinfection observed at waterwork I despite the lower DOC concentration. Higher residual free chlorine concentration in BC leads to a lower HOBr/HOCl ratio; thus, the impact of $Br^-$ ion on DBP formation is less relevant.

Chlorate

　　Chlorate concentration measured after BC ranged from 0.15 to 0.71 mg/L with a mean value of 0.42 mg/L at waterwork I, 0.38–1.1 mg/L (mean: 0.73 mg/L) at waterwork II, and 0.59–1.0 mg/L (mean: 0.76 mg/L) at waterwork III. The concentration in all but one sample exceeded the parametric value of the recast of the DWD (0.25 mg/L), and, in 42%, the less stringent value (0.70 mg/L), for exceptional situations even before disinfection. The mean additional chlorate concentration resulting from disinfection was considerably lower (0.034, 0.053, and 0.12 mg/L at sites I, II, and III, respectively). Chlorate is mainly produced in a disproportion reaction during the storage of hypochlorite solutions [28]. The higher chlorine dosage during BC introduces a larger amount of chlorate. The high SDs and wide concentration ranges reflect the differences between the applied hypochlorite solutions.

3.1.3. Bromine Substitution Factors

　　The species distribution of DBPs is almost as important as their concentration, since brominated compounds have higher cytotoxicity and genotoxicity [13,14]. The bromine substitution was significantly higher for each DBP class at waterwork I, where the $Br^-$ concentration of the raw water was the highest (Table 3). The formation of the brominated DBPs is more pronounced during disinfection, resulting in higher BSF values, than during BC. Results indicate that the main driver of BSF is the $Br^-$:free $Cl_2$ ratio, and not the actual bromide concentration. Thus, despite the lower residual free chlorine values, the adverse health effect of disinfection may be more severe.

**Table 3.** Calculated BSF (with standard deviation) of DBP classes at different chlorination processes.

| DBPs | Waterwork I | | Waterwork II | | Waterwork III | |
|---|---|---|---|---|---|---|
| | BC | Disinf. | BC | Disinf. | BC | Disinf. |
| THMs | 0.14 (0.024) | 0.72 (0.025) | 0.021 (0.006) | 0.15 (0.034) | 0.011 (0.006) | 0.13 (0.11) |
| DHAAs | 0.35 (0.15) | 0.71 (0.11) | 0.044 (0.030) | 0.35 (0.19) | 0.021 (0.018) | 0.14 (0.17) |
| THAAs | 0.093 (0.042) | 0.39 (0.17) | 0.027 (0.013) | 0.079 (0.014) | 0.011 (0.009) | 0.049 (0.037) |
| DHANs | 0.33 (0.040) | 0.88 (0.023) | 0.056 (0.013) | 0.14 (0.014) | 0.024 (0.020) | 0.057 (0.040) |

　　Generally, the BSF of THAAs was the smallest, which agrees with the findings of Obolensky and Singer, 2005, and Hua and Reckhow, 2012 [36,40]. The BSF of other DBP classes varies from site to site. Mostly, the BSF of disubstituted DBPs were higher than that of the trisubstituted ones.

### 3.1.4. Efficiency of GAC Filtration

GAC or BAC filters are always used after BC to reduce the concentration of the produced DBPs and to eliminate the excess free chlorine. The operational parameters of GAC filtration at the investigated waterworks are summarized in Table S2. The concentration of residual free chlorine was reduced below LOQ by GAC filtration during the entire sampling campaign.

The removal efficiencies for the DBP classes varied between sites and over time (Figure 3) Initially, the removal of THMs and DHAAs was the lowest (39–62% and 53–65%, respectively). The removal efficiency of THAAs was between 70 and 83%, whereas the highest values (above 75%) were measured for DHANs. The observed removal rates are in accordance with the Freundlich adsorption coefficient of the compounds: $K_{Chloroform} < K_{BDCM} < K_{DCAA} < K_{TCAA} < K_{DCAN}$ [41]. The correlation indicates that, initially after installation of the filters, DBPs are mainly eliminated by adsorption.

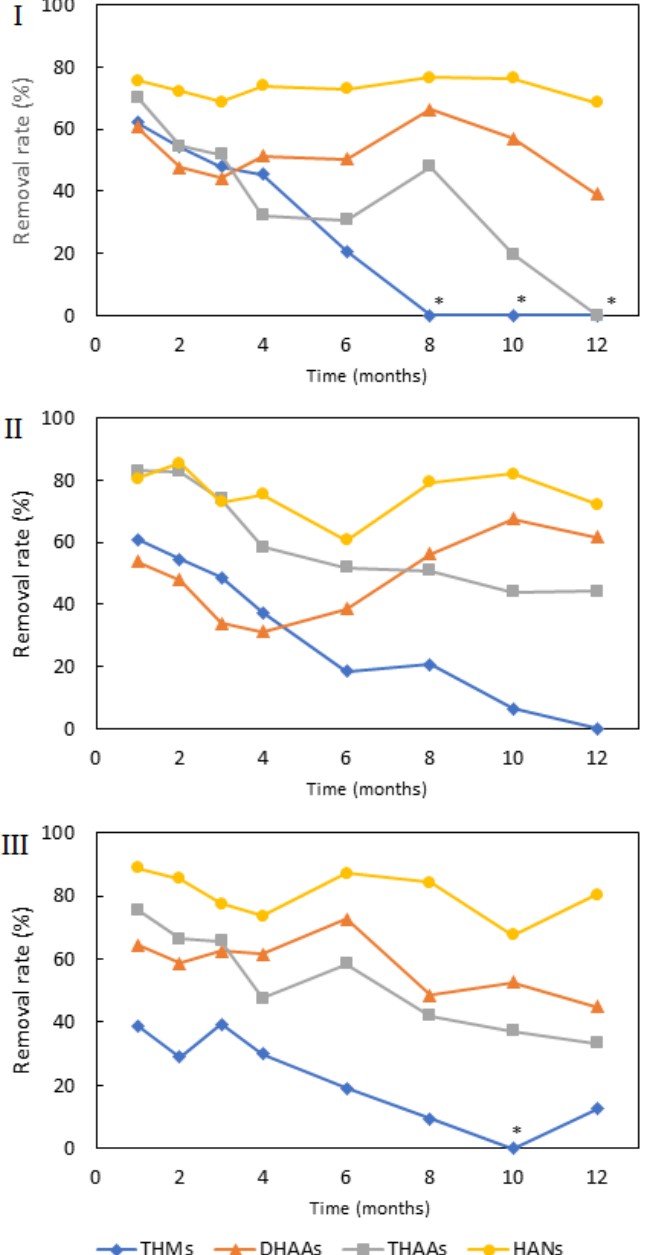

**Figure 3.** Removal rate of the GAC adsorbents at waterworks **I–III** (* Calculated removal rate was below 0%).

The adsorption capacity of GAC for THMs was exhausted rapidly, as reported earlier by Babi et al., 2007 and Kim and Kang, 2008 [42,43]. Removal rates were under 20 and 10% after 6 and 10 months, respectively (Figure 3). The lowest removal rates and fastest depletion were observed for chloroform, whereas brominated compounds were more readily removed, in accordance with Freundlich coefficients of the compounds: $K_{Chloroform} < K_{BDCM} < K_{DBCM} < K_{Bromoform}$ [41].

Biological removal of THMs is unlikely, due to their poor biodegradability [21]. On several occasions later in the sampling period, THM concentration was higher after GAC filtration than before (marked with * in Figure 3). The increase can be attributed either to the desorption of previously adsorbed compounds or the reaction of residual free chlorine with the absorbed organic components before its catalytic decomposition.

Removal efficiency of THAAs also decreased, from the initial 70–83 to 0–50% after one year, depending on the different operating conditions and loading rates of the sorbents. The removal of DHAAs (30–70%) did not change significantly during the sampling campaign. Thus, after several months of operation, the removal of DHAAs became more effective than THAAs (Figure 3). DCAA and TCAA are the predominant HAAs formed during BC. The adsorption coefficient of TCAA on activated carbon is higher [41], resulting in higher initial removal efficiencies. However, HAAs are biodegradable, DHAAs more readily than THAAs [44]. The change in the order of removal efficiencies reflects the increasing biological activity on the GAC sorbents. According to Wu and Xie, 2005, biological removal of TCAA requires longer contact time or higher water temperature than in DHAAs [45]. The low contact times at the investigated waterworks are likely to be the limiting factor of biological removal.

The highest removal rates were observed for HANs throughout the sampling period. Initially the concentration of HANs was reduced below LOQ at every site, due to the particularly high Freundlich coefficient of HANs [41]. At site I, the concentration of HANs remained under LOQ in the effluent during the entire sampling campaign, whereas at sites II and III, DCAN was detected in low concentrations (0.30–0.88 μg/L) 3–8 months after the installation. Nevertheless, the results indicate high removal efficiency of HANs even after one year. HANs are also biodegradable, so both GAC and BAC can be readily used for their removal.

Chlorate removal on the GAC cartridges was limited, on average 2.0, 4.6, and 23% at sites I, II, and III, respectively. Generally, GAC sorbents are used to eliminate organic compounds, such as taste and odor compounds [46], pesticides [47], or pharmaceutical residues [48] from waters. The adsorption of inorganic compounds on GAC is very limited. However, after some modification, it can be used for ionic compounds, e.g., also for heavy metal elimination [49,50]. Presumably, ions are removed through an ion exchange mechanism. Although Liu et al., 2017 reported some biologically active filtration processes for chlorate elimination [51], there is no economically feasible technology for reducing chlorate concentration; thus, it should be prevented from entering the treated water.

### 3.2. Bench-Scale Experiments

3.2.1. Breakpoint Chlorination

Bromide to Chlorine Ratio

The breakpoint reaction was complete in less than the 15 min contact time in every combination of bromide and free chlorine concentration, based on the measurements of residual ammonium and chlorine. The concentration of each DBP class increased with increasing initial Br$^-$ concentration (Figure 4). Between the lowest and highest Br$^-$ levels, an average increase of 47, 38, and 39% was observed for THMs, HAAs, and HANs, respectively. These results confirm our field study observations indicating that bromine containing oxidizing agents react faster with NOM than HOCl, forming more DBPs.

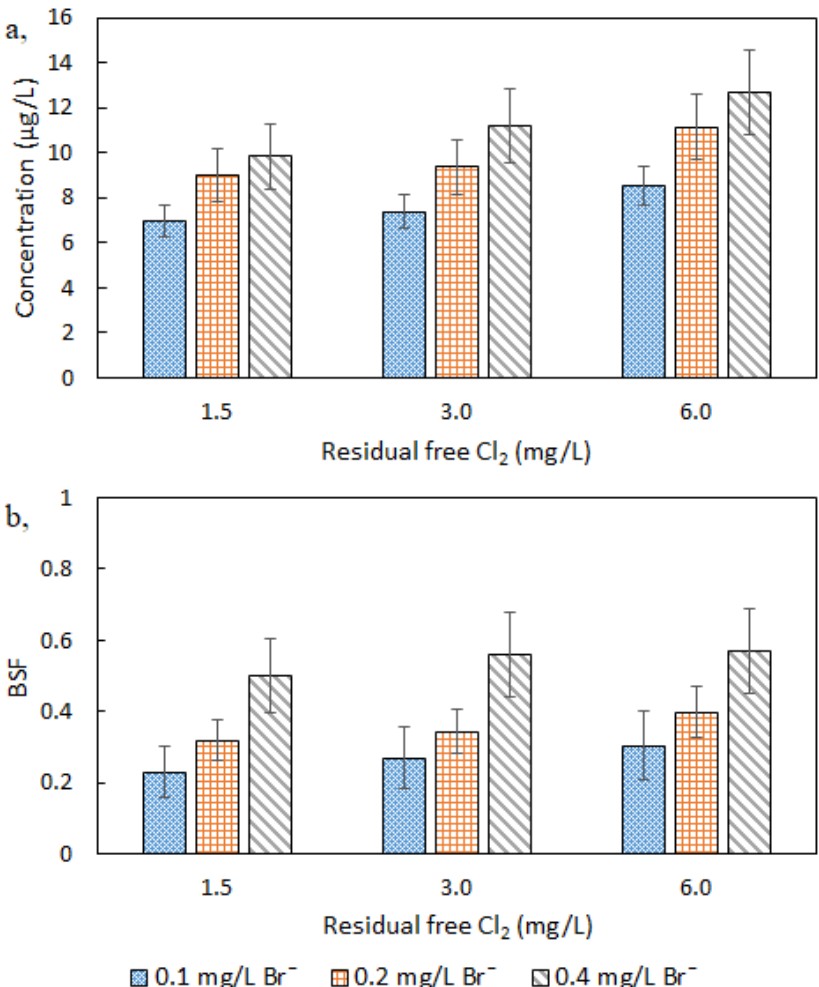

**Figure 4.** Effect of Br⁻ concentration and residual free $Cl_2$ in raw water on the (**a**) concentration and (**b**) distribution of THMs during BC.

The residual free chlorine concentration had a lower impact than the bromine concentration. Concentrations measured at the highest chlorine dosage were, on average, 25 and 26% higher for THMs and HAAs compared to the lowest dosage, whereas a consistent increase was not observed for HANs. Results suggest that chlorine dosage and residual free chlorine concentration has only a limited influence on DBP formation. Free chlorine is applied in great excess in BC and it is not the limiting factor of DBP formation. Reducing the chlorine dosage therefore is not an efficient option for controlling DBPs. Moreover, residual free chlorine concentration cannot be adjusted precisely because of the high required dose for breakpoint reaction, the uncertainty of hypochlorite solutions' concentration, and further potential chlorine consuming reactions.

BSF is also mainly influenced by bromide concentration. Between the lowest and highest Br⁻ level, BSF values increased by 0.27, 0.22, 0.25, and 0.33, on average, for THMs, DHAAs, THAAs, and DHANs respectively (Figure 4, Figures S1–S3). Increasing residual free chlorine concentration had only a minor impact, i.e., either a slight decrease (disubstituted DBPs, DHAAs, and DHANs) or a slight increase (trisubstituted DBP, THMs, and THAAs). The highest BSF values were observed for DHANs (range: 0.38–0.80) followed by DHAAs (0.30–0.57), THMs (0.23–0.57), and THAAs (0.03–0.32), similar to the observations of the field study.

Contact Time

The breakpoint reaction was complete in 12 min. Ammonium decreased below the LOQ, whereas the residual free chlorine concentration stabilized around 2.0 mg $Cl_2$/L after the breakpoint. The dominant DBPs were HAAs, which were produced mostly in the first 3 min to a final concentration of 15 and 20 µg/L (Figure S4). The concentration of THMs and HANs was much lower in the first minutes of the breakpoint reaction, but increased steadily during the observational period of 30 min up to 14 and 6.1 µg/L, respectively. Similar dynamics were observed previously in the case of disinfection [26]. The concentration of THMs and HANs continued to grow after the breakpoint by 70 and 35%, respectively, mainly due to the formation of brominated DBPs, DBCM, bromoform, and DBAN (Figure 5). Chlorinated DBPs (chloroform and DCAN) were generated immediately after hypochlorite addition, when free chlorine concentration is particularly high in the treated water. The initial reaction step (oxidation of ammonium to monochloramine) is fast ($k = 4.2 \times 10^6$ $M^{-1}$ $s^{-1}$) [52], but the total BC takes several minutes. Although the oxidation of $Br^-$ also occurs in a fast side reaction ($k = 1.32 \times 10^6$ $M^{-2}$ $s^{-1}$) [37], the ratio of HOBr/HOCl is still low in the first minutes. Thus, the bromine incorporation into DBPs is limited, and chlorinated DBPs are mainly formed. As the breakpoint reaction progresses, the HOBr/HOCl ratio increases, which promotes the formation of brominated DBPs. Accordingly, the BSFs also increased over time (Figure S6). After 3 min, BSFs for THMs, DHAAs, THAAs, and DHANs were 0.224, 0.596, 0.083, and 0.356, and these values increased to 0.555, 0.625, 0.233, and 0.648, respectively, after 30 min. Results indicate that the appropriate contact time is particularly important in controlling the formation of DBPs, especially the more toxic brominated ones. Excess breakpoint reagent should be removed immediately after the elimination of ammonium. Contact time adjustment can also defer the depletion of the GAC adsorbents.

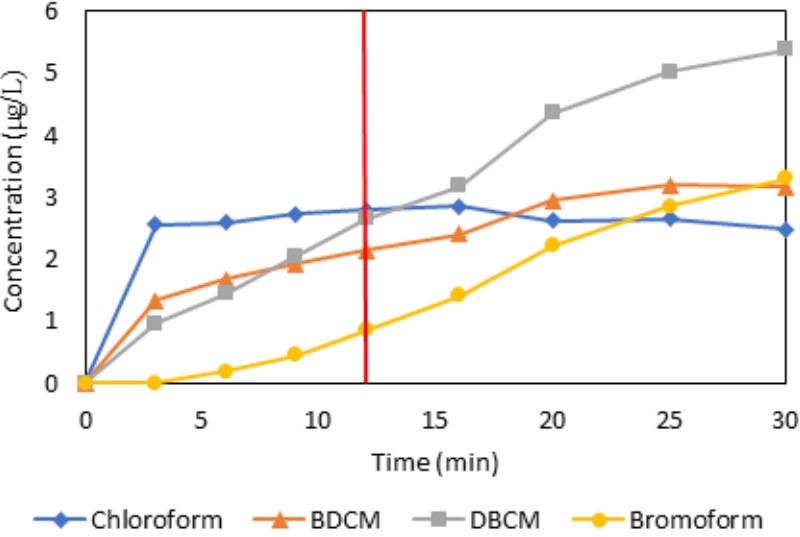

**Figure 5.** The concentration of individual THMs during BC (red line indicates the estimated time when breakpoint reaction is complete).

3.2.2. Disinfection

Bromide to Chlorine Ratio

In disinfection simulation experiments, the lower doses of chlorine (0.30 and 0.60 mg $Cl_2$/L) were eliminated in 24 h, whereas the highest (1.2 mg/L) was reduced by 40–60%. The dominant DBPs were THMs, generated in concentrations that were 2–4 times higher than those of HAAs. The concentrations of THMs, HAAs, and HANs increased at higher initial $Br^-$ levels (Figure 6, Figures S5–S7). The most significant increment was found in samples with the highest chlorine dosage, where the concentrations of THMs, HAAs, and HANs increased by 90, 53, and 74%, respectively. The effect of higher chlorine

dosage was even more significant. The concentration of the formed DBPs was 4–10 times higher at the highest chlorine dose (1.2 mg $Cl_2$/L) compared to the samples containing the lowest free chlorine level (0.30 mg $Cl_2$/L).

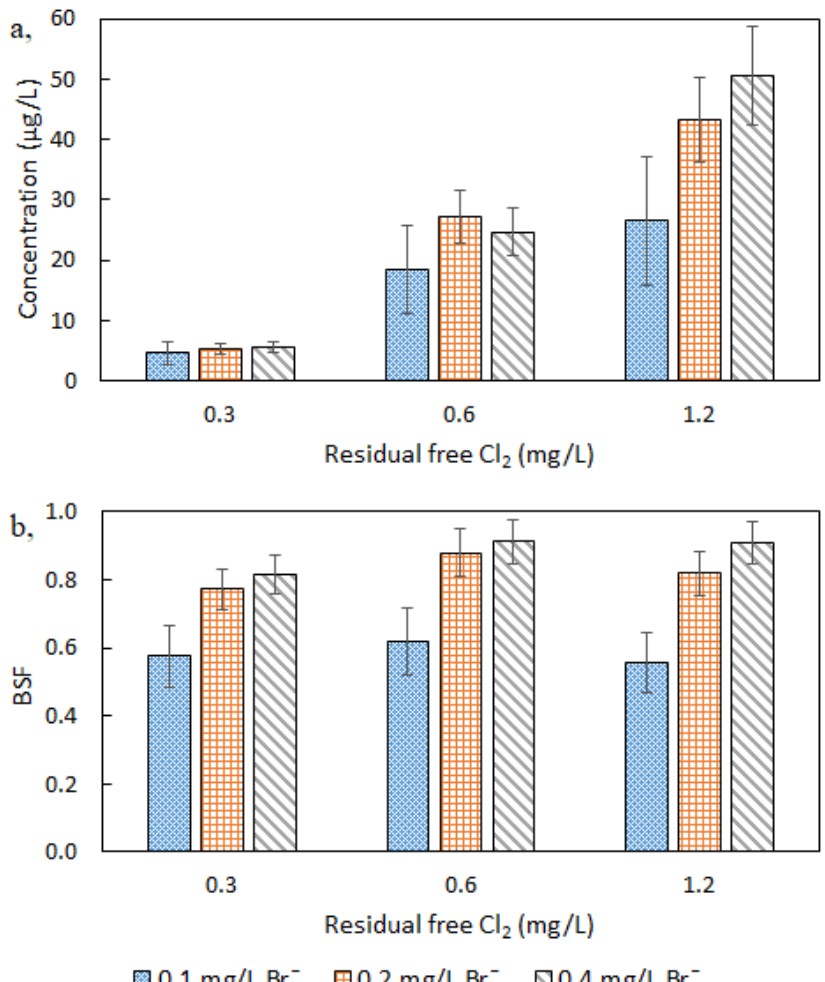

**Figure 6.** The effect of $Br^-$ concentration and residual free $Cl_2$ in raw water on the (**a**) concentration and (**b**) distribution of THMs at bench-scale disinfection experiments.

The BSF values for THMs, DHAAs, THAAs, and HANs were 0.55–0.91, 0.63–0.93, 0.47–1.0, and 0.80–1.0, respectively (Figure 6, Figures S5–S7). In several reactions, only brominated DBPs were formed in detectable concentrations (exceeding the LOQ). Generally, the higher values were observed in samples with higher $Br^-$ levels. BSF was increased by 0.15–0.46 between the lowest and highest $Br^-$ level on average. The effect of chlorine dosage less well defined. The BSF values varied within the margin of error in the samples with the same bromide but different free chlorine levels.

### 3.2.3. Comparison of Breakpoint Chlorination and Disinfection

The formation of HAAs was dominant over THMs and HANs during breakpoint chlorination until the breakpoint reaction was complete, whereas THMs were formed in the highest concentration during disinfection. Generally, 2–4 times higher concentrations were observed for THMs than for HAAs. Previous studies reported that the ratio of certain DBP classes is mainly affected by the pH and the composition of DOC [53]. In the present study, the applied DOC and the pH was the same in all bench-scale experiments; thus, the observed differences cannot be attributed to these factors. THAAs can degrade through hydrolysis or decarboxylation to THMs, but not in the timeframe of the test period (1 day) [54]. The relatively high concentration of HAAs during BC can be explained by the

rapid formation of DHAAs reported by Hua and Reckhow, 2008, but further investigations are necessary to better understand this phenomenon [26].

The effect of initial $Br^-$ level on the concentration of DBPs was similar in the BC and disinfection experiments. The impact of chlorine dosage is different: whereas in disinfection experiments residual free chlorine is the limiting factor of the DBP formation, during BC it has practically no effect. Above the breakpoint, the concentration of DBPs cannot be controlled effectively by the adjustment of residual free chlorine, but rather by raw water composition (DOC, $Br^-$ concentration) and physico-chemical parameters of the water [55].

Distribution of DBPs is determined by the concentration of $Br^-$ during both BC and disinfection. BSFs are not affected significantly by the chlorine dosage in either treatment. Nevertheless, the BSF values during disinfection experiments were particularly high compared to breakpoint experiments, even at similar $Br^-/Cl_2$ ratios. For example, the BSFs at BC with 0.20 mg/L $Br^-$ and 1.2 mg/L free $Cl_2$ concentration were 0.318, 0.438, and 0.608 for THMs, DHAAs, and DHANs, respectively. During disinfection with 0.20 mg/L $Br^-$ and 1.5 mg/L free $Cl_2$ concentration, it was much higher (0.819, 0.900, and 0.923, respectively). This phenomenon is probably due to the difference in the contact time (minutes vs. hours/days). As was observed in the contact time experiment, in the first minutes after the reagent addition, the $Br^-/Cl_2$ ratio was much lower due to the presence of unreacted free $Cl_2$, and the formation of chlorinated DBPs is more likely. Although distribution of DBPs generated after the breakpoint may be similar to that observed during disinfection, the DBPs formed in the initial reaction result in lower BSF values.

## 4. Conclusions

Both field sampling and bench-scale experiments indicated significant differences between BC and disinfection in the formation of DBPs. The observations of field sampling and bench-scale experiments were in good agreement regarding the key factors of DBP formation. Due to the harmful health effect of organic DBPs, it is important for water supply operators to be aware of the characteristics of each process for appropriate control of by-products.

At BC, the formation of DBPs is limited by contact time and DOC concentration of raw water. Chlorine dosage has only a limited impact on the produced DBPs. During disinfection, DBP formation is limited primarily by the residual free chlorine. Disinfection is more likely to generate brominated DBPs from identical raw water. Since brominated THMs, HAAs, and HANs present a higher risk to health than chlorinated ones, adverse health effects of disinfection without the control of DBPs may be similar to those of BC, despite the considerably higher chlorine dosage of the latter.

The new parameters introduced in the recast European regulation present a novel situation for the water suppliers. The results of the field study indicate that non-compliance with the HAA parametric value is unlikely in Hungarian water supplies, including those applying BC. However, complying with the chlorate parametric value will be a challenge for water supplies using hypochlorite solutions as the breakpoint reagent. Appropriate storage conditions (cool, protected from sunlight) can reduce the formation of chlorate, but the final solution for its elimination is the use of chlorine gas or electrochemically in situ produced hypochlorite for BC. The use of chlorine-free techniques for ammonium elimination, such as biological oxidation, is also an option.

GAC filters can be useful for the removal of DBPs formed during BC. Unfortunately, the lowest removal rates and the fastest depletion are expected for the chlorinated THMs, which are predominantly produced during BC. An effective regeneration or replacement of sorbents are necessary after the depletion (6–8 months) of GAC. Possible options are thermal regeneration [56] and chemical regeneration [57]. Recent studies indicate that mixtures (e.g., NaOH/ethanol) are more effective in chemical regeneration [58]. On the other hand, the biological activity of the GAC filters increases after several months of operation, which improves the removal of HAAs and HANs. At this stage, activated carbon cartridges are operating more like a biologically activated carbon (BAC) filter than a GAC. As a result,

the emergence of harmful microorganisms in the filtered water is more likely; thus, proper disinfection, as a final step in the treatment process, is particularly relevant. There are multiple commercially available options for DBP elimination, such as membrane filtration (reverse osmosis, nanofiltration) [59] or different oxidation processes [60]. Although the efficiency of the above methods for DBP removal may be greater, GAC and BAC filters provide the optimal combination of cost efficiency, efficacy, and ease of operation.

The results of the current study will be utilized in practice of the operation of water supplies using BC in Hungary to optimize water treatment parameters and operational practices, and thus minimize the health risk of DBPs via drinking water consumption.

Further research is necessary to better understand the rapid formation of DHAAs during BC. Decomposition of biodegradable HAAs and HANs may occur if the biological activity of the water increased, i.e., due to biofilm formation on the GAC filters or in the distribution system. Preferential formation or decomposition of DBPs under different conditions in water distribution and their impact on human health requires further study.

**Supplementary Materials:** The following supporting information can be downloaded at: https://www.mdpi.com/article/10.3390/w14091372/s1. Table S1: Main operational differences between chlorination technologies used for disinfection and breakpoint chlorination; Table S2: Technology parameters of GAC adsorption at the investigated waterworks; Figure S1: The effect of raw water $Br^-$ and res. free $Cl_2$ on the (a) concentration and (b) distribution of DHAAs during BC; Figure S2: The effect of raw water $Br^-$ and res. free $Cl_2$ on the (a) concentration and (b) distribution of THAAs during BC; Figure S3: The effect of raw water $Br^-$ and res. free $Cl_2$ on the (a) concentration and (b) distribution of DHANs during BC; Figure S4: The (a) concentration and (b) distribution of DBPs in time during bench-scale BC experiments (red line indicates the estimated time when breakpoint reaction is complete); Figure S5: The effect of raw water $Br^-$ and residual free $Cl_2$ on the (a) concentration and (b) distribution of DHAAs in bench-scale disinfection experiments; Figure S6: The effect of raw water $Br^-$ and residual free $Cl_2$ on the (a) concentration and (b) distribution of THAAs in bench-scale disinfection experiments; Figure S7: The effect of raw water $Br^-$ and residual free $Cl_2$ on the (a) concentration and (b) distribution of DHANs in bench-scale disinfection experiments.

**Author Contributions:** Conceptualization, D.S., G.Z. and M.V.; Investigation, D.S. and J.B.; Methodology, D.S. and J.B. data curation, D.S. and J.B.; writing—original draft preparation, D.S.; writing—review and editing, M.V.; visualization, D.S.; supervision, G.Z. and M.V.; All authors have read and agreed to the published version of the manuscript.

**Funding:** This research received no external funding.

**Institutional Review Board Statement:** Not applicable.

**Informed Consent Statement:** Not applicable.

**Data Availability Statement:** All data presented in this study are contained within the article and the Supplementary Materials.

**Acknowledgments:** We acknowledge the contribution and help of Hungarian waterworks. The authors are grateful to thesis workers and colleagues for all their help in performing some of the analytical measurements.

**Conflicts of Interest:** The authors declare no conflict of interest.

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
