# Peer review of "Comparison of Disinfection By-Product Formation and Distribution during Breakpoint Chlorination and Chlorine-Based Disinfection in Drinking Water"

_water, doi:10.3390/w14091372_

Round 1
Reviewer 1 Report
Water-1664863 compares the formation and the distribution of organic and inorganic DBPs (THMs, HAAs, HANs, and chlorate) during chlorination breakpoint and chlorine-based disinfection. The authors designed two sets of experiments, which included a field investigation and a bench-scale experiment. Besides this, the impact of the aging a GAC adsorbents on the distribution of DBPs formation was also studied. The structure of this paper is well-organized, and I am interested in this topic. I recommend to accept this manuscript after a minor revision. Here are some specific suggestions.
- Figure S1 should be added into the main text, providing convenience to understand the difference between breakpoint chlorination and disinfection.
- The introduction has too many paragraphs that is fails to address the necessity of this study. I recommend to combine the risks of DBPs into one paragraph, and strengthen the comparison of break chlorination and disinfection in terms of their different technological processes. Also, some up-to-date studies should be cited.
- In section 2.3, more details of the analytical methods of DBPs should be included, but not just list the standard methods in Table 1.
- The * labels in Figure 1 are quite puzzled, because they are not located at any columns. Also, is there any significant difference in Figure 3?
- GAC would be inevitably converted into biologically activated carbon (BAC) when they are exhausted. Since the BAC can also lead to the degradation of some DBPs, it sounds like a trade-off. So do you have some discussion or suggestion on the regeneration of GAC?

Reviewer 2 Report
Water 1664863
Comparison of disinfection by-product formation and distribution during breakpoint chlorination and chlorine-based disinfection
The manuscript presents information on the formation of disinfection by-product during the treatment of groundwater. Two sets of experimental work were done, namely field and lab-scale experiments to assess the difference in the amount and distribution of these products due to breakpoint operation and disinfection using sodium hypoclorite. The field experiments were conducted in three drinking water treatment plants where 5 sampling points were used at each plant. The lab scale experiments were conducted on simulated water to study the effect of chloride and bromide content on the formation of the DBP due to breakpoint and disinfection. The topic is suitable for publication in water, and the manuscript is recommended for publication after revision to clarify the field experimental work and its corresponding results and discussion.
in line 97, pls indicate the water treatment plants as (I, II, III, or any other numeral order) to avoid any confusion with the sampling points (a, b, c). pls apply the change throughout the text
in line 201 pls provide information about the resident time at the three sites
in line 303 pls revise “The impact of residual free chlorine concentration was less pronounced”
for each ref cited at the end of the text, the format should be the ref followed by followstop “[MMM]. ”
the list of ref needs to be updated with more recent references
Reviewer 3 Report
The manuscript is accepted for the publication after following changes
- The manuscript is well written and the concept proven is very helpful for the readers.
- Discuss the role of heteroatom doped high surface area carbon material for adsorption. Discuss referring to these papers: Journal of Environmental Chemical Engineering, 2021, 9, 106656; ACS Biomaterials Science & Engineering, 2020, 6, 5527–5537.
- The author stated that the biological activity of the filter started to increase after 3-6 months of operation. Why is it so?
- The filter works well for the DBPs. Will it work for other heavy metals or other contaminants in drinking water? Explain.
- The removal efficiency provided in Figure 2 is not showing a proper trend of either increasing or decreasing after some months for different contaminants. Why?
- What is the state art of GAC and BAC from the commercial point of view? Compare the data with other commercially available filters.
Reviewer 4 Report
Dear authors,
Thank you very much for inviting me to review the manuscript "Comparison of disinfection by-product formation and distribution during breakpoint chlorination and chlorine-based disinfection”. The manuscript is very interesting, the research is well conducted, the results are adequately described and analyzed, and the experimental method is accurate, however there are some issues that can be addressed to improve the overall impact:
- Title:
- Present your discovery more clearly, provide more attractive and groundbreaking claim.
- Abstract
- Define the abbreviations when used for the first time (such as EU, GAC…)
- Introduction:
- The references must go in the cited paragraph, always before the end point. Manuscript authors should review the entire document and correct this error.
-The authors need to be expanded further to cover all the critical aspects of the topic. It is suggested to provides the consequences and adverse effects of disinfection by-products (DBPs) on health concern.
- More intensely compare your findings with existing literature, refer to methods used in developing countries and refer to papers "Effecting Partial Elimination of Isocyanuric Acid from Swimming Pool Water Systems" https://doi.org/10.3390/w11040712 .
- The authors should clarify that only active chlorine (mainly composed by HOCl and ClO-) is effective for disinfection.
- Build your research hypothesis more clearly (straightforward and groundbreaking claim that is confirmable or refutable) at the end of the Introduction chapter, justify the urgency of its investigation from drinking water treatment station point of view.
- Materials and Methods:
- All measurement international standard methods must be cited in the references (Table 1).
- Was the pH, hardness, alkalinity and temperature altered in the synthetic raw water to simulate drinking water?
- The equations that were used for calculated dihalogenated acetic acids (DHAAs), trihalogenated acetic acids (THAAs) and dihalogenated acetonitriles (DHANs) must be appear in the manuscript.
- Table 2 must be improved, the mean and SD of raw water must appear in separate columns for each of the investigated waterworks A, B and C.
- Conclusions:
- Lines 419 – 421: I suggest making this sentence clearer.
- Can it be problematic for treated water that biological activity of GAC filters increases over time? Used GAC filters can generate microorganisms in drinking water that are harmful to human health
- Indicate direction for future research.
- Others:
- It would be convenient for the manuscript to be reviewed by a native English speaker so that any grammatical errors it may contain are corrected.
Round 2
Reviewer 4 Report
Dear Authors,
This reviewer commends the authors efforts in addressing the comments. The new version of the manuscript has been improved according to the comments and corrections suggested by the reviewers.